# First Report of Paralytic Rabies in a Lowland Tapir (*Tapirus terrestris*) in Argentina

**DOI:** 10.3390/v17040570

**Published:** 2025-04-15

**Authors:** Matías Castillo Giraudo, María Marcela Orozco, Marcelo Juan Zabalza, Leonardo Minatel, Laura Patricia Novaro, Gabriela Alejandra Centurión, Marcos Adolfo Fabeiro, Luciano Coppola, Vanina Daniela Marchione, María Carolina Artuso, Pablo Daniel Aon, Susana Elida Russo

**Affiliations:** 1Estación de Animales Silvestres Guaycolec, J.M. Uriburu 1513, Formosa 3600, Argentina; matigiraudo12@gmail.com; 2Facultad de Ciencias Exactas y Naturales, Universidad de Buenos Aires, Instituto de Ecología, Genética y Evolución de Buenos Aires, Consejo Nacional de Investigaciones Científicas y Técnicas (IEGEBA-CONICET), Av. Intendente Güiraldes 2160 Guiraldes, Ciudad Autónoma de Buenos Aires C1428EGA, Argentina; 3Dirección de Laboratorio Animal (DLA), DGLyCT, SENASA, Talcahuano 1660, Martínez B1640CZT, Argentina; mzabalza@senasa.gob.ar (M.J.Z.); lnovaro@senasa.gob.ar (L.P.N.); gcenturion@senasa.gob.ar (G.A.C.); mfabeiro@senasa.gob.ar (M.A.F.); lcoppola@senasa.gob.ar (L.C.); vmarchione@senasa.gob.ar (V.D.M.); mcartuso@senasa.gob.ar (M.C.A.); srusso@senasa.gob.ar (S.E.R.); 4Facultad de Ciencias Veterinarias, Universidad de Buenos Aires, Chorroarín 280, Ciudad Autónoma de Buenos Aires C1427CWO, Argentina; lminatel@fvet.uba.ar; 5Asociación Civil Grupo Silvestre, Ciudad Autónoma de Buenos Aires C1217ABK, Argentina; pabloaon@gmail.com

**Keywords:** rabies, *Tapirus terrestris*, paralytic rabies, wildlife surveillance, zoonosis, antigenic variant 3, *Desmodus rotundus*, one health

## Abstract

As a significant zoonotic disease, rabies poses substantial economic challenges for the livestock sector, highlighting the need for effective wildlife monitoring as part of a One Health approach. This study documents the first case of paralytic rabies in a lowland tapir (*Tapirus terrestris*) at the Guaycolec Wildlife Station in Formosa, Argentina. The 12-year-old male tapir exhibited neurological symptoms, including limb paralysis and dysphagia, leading to its death. The rabies virus was confirmed through direct immunofluorescence, virus isolation in BHK-21 cells, and molecular diagnostics via real-time RT-PCR and conventional PCR. Antigenic variant 3, associated with *Desmodus rotundus*, was identified. Histopathological examination revealed non-suppurative encephalitis with lymphocytic perivascular cuffs, neuronal vacuolization, and acidophilic intracytoplasmic inclusion bodies in the grey matter. This case underscores the importance of expanded surveillance for non-traditional hosts, as it demonstrates the potential for rabies transmission in changing environments. The findings highlight the need to maintain epidemiological surveillance systems at the wildlife–livestock–human interface and to develop targeted control strategies to mitigate the spread of rabies, particularly in areas where vampire bat populations are subject to anthropogenic pressures. Comprehensive monitoring and early detection are essential for effective rabies management in both wildlife and urban contexts.

## 1. Introduction

Rabies is a zoonotic disease caused by the rabies virus (Lyssavirus, Rhabdoviridae), which remains a significant global public health concern, accounting for approximately 59,000 human deaths annually and over 3.7 million disability-adjusted life years (DALYs) [1]. The disease affects mammals across all continents except Antarctica and is characterized by two primary epidemiological cycles: urban rabies, where domestic dogs serve as the main reservoir and transmitter, and sylvatic rabies, involving a variety of wild and domestic species [2,3]. Rabies also presents a significant conservation challenge [4], with the rabies virus having been detected in over 190 species of wild mammals, including several that are threatened. In particular, the spread from domestic dogs has been implicated in the decline of populations of species such as the Ethiopian wolf (*Canis simensis*) and the African wild dog (*Lycaon pictus*) [5,6].

In Latin America, epidemiological surveillance remains essential to prevent outbreaks and achieve the global goal of eliminating dog-mediated rabies by 2030 [7]. However, paralytic rabies, primarily transmitted by the common vampire bat (*Desmodus rotundus*), remains a significant challenge for livestock, which serves as both a reservoir and transmitter of the virus to domestic animals, leading to considerable economic losses [8,9]. Between 1970 and 2023, the region documented an average of 450 outbreaks annually, with Brazil, Colombia, Peru, and Mexico being the most affected countries. The increase in outbreak frequency from 2000 to 2020, compared to previous decades, underscores the escalating concern regarding the virus’s spread, particularly as cross-species transmission events have intensified since 2002. While the recent decline in outbreak size and cattle mortality rates [9,10,11] suggests progress, sustained vigilance and targeted intervention strategies are essential to mitigate the health threat posed by rabies transmitted by *Desmodus*. Effectively addressing these challenges is vital for enhancing ecosystem health and supporting the socioeconomic well-being of affected communities.

In Argentina, rabies cases have been exclusively associated with the classical rabies virus genotype 1, comprising five identified antigenic variants (AgV1–4, AgV6), each linked to different hosts [12]. Specifically, AgV1 and AgV2 are associated with rabies in domestic dogs, while AgV3 is linked to vampire bats. AgV4 has been detected in non-hematophagous and insectivorous bats, and AgV6 has been isolated from insectivorous bats [13]. Paralytic rabies was first recorded in 1928, entering through the provinces of Corrientes and Formosa from Paraguay, and has since spread north of parallel 31° S and east of meridian 66° W. Between 1970 and 2023, outbreaks varied significantly, ranging from 1 to 1360 cases per event, with an average size of 14.6 cases and an annual incidence of 93.1 cases, indicating a lower outbreak frequency compared to tropical countries such as Brazil and Peru [10]. The impact of rabies extends beyond livestock, affecting cattle, horses, humans, and occasionally wildlife, with rabies in wildlife predominantly associated with vampire bats [14,15,16,17]. It has been reported that during rabies outbreaks in livestock, the virus has been detected in several wild species, including marsh deer (*Blastocerus dichotomus*), red brocket deer (*Mazama americana*), and capybara (*Hydrochoerus hydrochaeris*) [16]. Although insectivorous bats are not major vectors of rabies, they have been implicated in viral circulation, showing positivity rates ranging from 3.1% to 5.4% in species such as *Tadarida brasiliensis, Myotis* spp., *Eptesicus* spp., *Histiotus montanus*, *Lasiurus blossevillii*, and *Lasiurus cinereus* [18]. Genetic analyses have identified six distinct viral lineages, some of which co-circulate with variants found in other American countries, and interspecies transmission, particularly among *Lasiurus* species, has been documented.

In northern Argentina, the lowland tapir (*Tapirus terrestris*), classified as a vulnerable species, plays a key ecological role in maintaining forest structure and facilitating seed dispersal throughout its habitat. It is present in the northern Yungas, the Paranaense Forest, and the northern Chaco woodlands [19] (Figure 1). The species faces numerous threats, including habitat loss and degradation, fragmentation, hunting, and conflicts with human activities, which have contributed to declining populations [20]. While various pathogens, such as *Theileria* spp. [21,22], *Bartonella* spp. [23], *Borrelia* spp. [24], *Leptospira interrogans*, *Salmonella* spp., and mycobacteria from the *Mycobacterium avium* complex (MAC) and *Mycobacterium tuberculosis* complex (MTC), have been identified in tapirs [25,26,27], cases of rabies are exceedingly rare. The only previously documented case of rabies in tapirs occurred in São Paulo, Brazil, where the animal experienced progressive neurological deterioration leading to its death [28]. In this study, we present the first confirmed case of rabies in a lowland tapir in Argentina, contributing to the understanding of rabies epidemiology in this vulnerable species and highlighting the need for increased rabies surveillance and conservation efforts.

## 2. Materials and Methods

### 2.1. Study Subject

The subject of this study was a lowland tapir that had been part of the permanent stock at the Guaycolec Wildlife Station in Formosa Province, Argentina, since 2014. At the time of the study, it was housed in a 3-hectare enclosure, separated from a female and her offspring. The enclosure was located in a forested area with two internal lagoons. Formosa Province is located in northeastern Argentina, bordered by Paraguay to the north and the Brazilian state of Mato Grosso do Sul to the northeast. The region is characterized by a warm climate, with maximum temperatures ranging from 21.1 °C in July to 32.5 °C in January. Relative humidity remains high throughout the year, fluctuating between 64% and 78%, and precipitation peaks in April, with an average of 171 mm [29]. The driest months are July and August. The Guaycolec Wildlife Rescue Center is located to the east of the province, at coordinates 25°58′55.1″ S, 58°09′42.4″ W (Figure 1). The animal had been voluntarily surrendered by a family from Potrero Norte, Formosa, after being found in the wild and raised in captivity for two years. Due to human imprinting, reintroduction to its natural habitat was deemed unfeasible, and the animal was maintained in captivity for breeding purposes. The tapir successfully reproduced with a female, resulting in two offspring in 2021 and 2023.

### 2.2. Clinical Evaluation

A complete anamnesis was conducted, collecting data on age, sex, medical history, symptom onset and progression, environment, diet, and potential exposure to toxins or infectious agents. A general physical examination was performed, assessing behavior, posture, and spontaneous movements, followed by a systematic evaluation of all organ systems. The neurological examination included assessments of mental status, gait and posture, cranial nerve function, spinal reflexes, and sensory responses. All information was recorded in the medical history.

### 2.3. Sampling Procedures

Upon the onset of progressive neurological symptoms and subsequent death, a full necropsy was performed. A macroscopic evaluation of all organs was conducted, and the entire brain was extracted and refrigerated for laboratory analysis. Blood samples were collected via cardiac puncture, with aliquots of 0.5–1 mL obtained. Tissue samples were stored in an ultra-freezer at −80 °C. Subsequently, some samples were thawed and fixed in 10% buffered formalin.

### 2.4. Laboratory Diagnosis

The suspected rabies samples were submitted from the local Formosa office to the Department of Rabies and Small Animal Diseases of the National Animal Health and Food Safety Service (SENASA). Brain tissue samples were collected for rabies diagnosis. Direct immunofluorescence (DIF) was performed using a commercial anti-nucleocapsid rabies conjugate (BIO-RAD Laboratories, Hercules, CA, USA). Viral isolation was conducted both in vivo and in vitro. A 20% (*w*/*v*) suspension in phosphate-buffered saline was prepared, which was used to infect BHK-21 clone 13 cell lines and intracerebrally to inoculate 10 albino mice (11–14 g) and 16 suckling mice.

Inoculated animals were observed for 28 days. Those exhibiting neurological symptoms consistent with rabies were euthanized, and their brains were extracted for indirect immunofluorescence (IIF) using a reduced panel of eight monoclonal antibodies provided by the CDC (Atlanta, GA, USA).

### 2.5. Molecular Diagnosis and Sequencing

Viral RNA was extracted from the samples as well as from mouse brain tissue used as a rabies-positive control, using the QIAamp Viral RNA Mini Kit (QIAGEN, Hilden, Germany).

For conventional PCR and subsequent nucleotide sequencing, the following primers were used: FW RAB 3S (5′ GGT CAY GTI TTC AAY CTC ATY CAC TT 3′) and RV 304 (5′ TTG ACG AAG ATC TTG GCT CAT 3′) [30,31]. PCR reactions were prepared in a final volume of 25 µL, comprising 20 µL of reaction mix and 5 µL of template sample. The reaction mix included 10 µL of nuclease-free bidistilled water, 5 µL of 5X buffer, 1 µL of 10 mM dNTP mix, 1.5 µL of forward primer 3S (10 µM), 1.5 µL of reverse primer 304 (10 µM), and 1 µL of Qiagen enzyme.

Conventional PCR was performed under the following thermal cycling conditions: an initial reverse transcription step at 50 °C for 30 min, followed by an initial denaturation at 94 °C for 15 min. This was followed by 40 cycles of denaturation at 94 °C for 45 s, annealing at 50 °C for 45 s, and extension at 72 °C for 1 min. A final extension step at 72 °C for 10 min was included to ensure complete amplification.

Real-time RT-PCR was conducted using the AgPath-ID One-Step RT-PCR Kit (Applied Biosystems, Foster City, CA, USA) on a 7500 Real-Time PCR System (Applied Biosystems, Foster City, CA, USA), following the LN34 Lyssavirus protocol [32] established by the CDC, Atlanta, GA, USA.

PCR products were purified using the Qiaex II Gel Extraction Kit (QIAGEN, Hilden, Germany) and sequenced with the BigDye Terminator v3.1 Cycle Sequencing Kit (Applied Biosystems, Foster City, CA, USA) on an ABI 3500 Genetic Analyzer (Applied Biosystems, Foster City, CA, USA). Sequence analysis and alignment were performed using BioEdit v7.2.5. Phylogenetic trees were constructed using the Molecular Evolutionary Genetics Analysis (MEGA) software version 11, employing the maximum likelihood method based on Kimura’s two-parameter model, as recommended by the software. To construct the phylogenetic tree, previously characterized sequences from different hosts (bovine, horse, fox, vampire bat, human, dogs, etc.) in Argentina, Chile, and Brazil, representing all variants, were used. These sequences originate from the database of the National Reference Laboratories for Rabies Diagnoses. In addition, a European sequence was included as an outgroup to root the tree. All sequences were manually aligned using BioEdit software.

### 2.6. Histopathological Diagnosis

Tissue samples fixed in 10% buffered formalin underwent standard histopathological processing [33]. Samples were embedded in paraffin wax, and 5-μm sections were obtained. These sections were stained with hematoxylin and eosin for microscopic examination, allowing for the identification and description of lesions.

## 3. Results

### 3.1. Clinical Findings

On April 2024, on the 20th, a 12-year-old male lowland tapir was discovered in lateral recumbency, exhibiting paddling movements on the ground (Figure 2). The animal displayed loss of consciousness and aggressive behavior, attempting to bite. Its mobility was limited to forward and backward limb movements, accompanied by a slight lateral head tilt. A vertical nystagmus was observed, which intensified with positional changes.

Initial vital signs included a heart rate of 70 beats per minute, which later decreased to 50 beats per minute, a respiratory rate of 30 breaths per minute, and a body temperature of 31 °C. During the first 48 h, a slight improvement was noted, characterized by the disappearance of nystagmus and a positive pupillary response to light. Paddling movements diminished and occurred only voluntarily. Although the aggressive signs subsided, lingual and laryngeal paralysis were detected during feeding attempts. By 96 h post-event, the symptoms progressed to complete limb paralysis, with continuous dorsal head movement, ultimately leading to a fatal outcome. Clinical findings included limb, lingual, and laryngeal paralysis, dysphagia, nystagmus, and aggression.

### 3.2. Necropsy Findings

A complete necropsy was performed, with samples collected from all organs, including the entire brain. Gross examination of the brain revealed congestion and mild edema, with no macroscopic lesions suggestive of trauma or hemorrhage. Other organs appeared grossly normal, with no significant pathological findings.

### 3.3. Laboratory Analysis

#### 3.3.1. Rabies Diagnosis

Imprint smears from various brain regions tested positive for the rabies virus using the direct immunofluorescence (DIF) technique (Figure 3A). Viral isolation was successfully conducted in BHK-21 clone 13 cells. Given that the virus does not produce cytopathic effects, DIF staining was employed, yielding positive results. Mice inoculated with neurologically symptomatic material were euthanized, and their brains were harvested for further analysis. Imprint smears were subsequently stained using the indirect immunofluorescence (IFI) technique with a reduced panel of monoclonal antibodies, confirming the presence of antigenic variant 3, associated with *Desmodus rotundus*, which was designated as RABV AC153 TAPIR.

#### 3.3.2. Molecular Diagnosis and Sequencing

Three brain samples were analyzed for rabies detection using primers 3S and 304 via conventional PCR, yielding the expected 550-bp amplicon in all RT-PCR-positive samples. Two of them, AC 153 original and passage, corresponded to the tapir protocol sample. The third one corresponded to a positive bovine sample.

The phylogenetic analysis with a reference database confirmed that the tapir sample (RABV AC153 TAPIR in red) was classified within variant 3 and grouped with genetically related sequences (Figure 4). Both genetic and antigenic characterization techniques consistently identified the sample as variant 3 (Figure 3 and Figure 4).

### 3.4. Histopathological Evaluation

Several small brain fragments were subjected to histopathological analysis. In the gray matter, blood vessels exhibited perivascular lymphocytic cuffs. In the most affected areas, mild diffuse gliosis was noted, while other regions displayed focal gliosis. Neuronal cytoplasm showed slight vacuolization, with displacement of Nissl substance. Intracytoplasmic acidophilic inclusion bodies were identified in multiple neurons, some of which were located in gray matter regions without signs of inflammation. The morphological diagnosis was diffuse non-suppurative encephalitis, severe, acute, with acidophilic intracytoplasmic inclusion bodies (Figure 5).

## 4. Discussion

This study presents the first confirmed case of rabies in a lowland tapir in Argentina, representing the second documented occurrence of the disease in the region, following a previous case in São Paulo, Brazil. The clinical signs and histopathological findings observed in the Argentine tapir closely resemble those reported in the Brazilian case [28]. Both tapirs exhibited neurological symptoms, including progressive paralysis and lateral recumbency, with a rapid deterioration of their condition after an initial phase of apparent improvement. Histopathological analyses in both cases revealed non-suppurative encephalitis characterized by perivascular lymphocytic cuffs, gliosis, and eosinophilic intracytoplasmic inclusions indicative of rabies virus infection.

In our study, the identification of the rabies virus and its antigenic variant 3 was achieved through a combination of diagnostic techniques, including DIF, IIF with a reduced panel of monoclonal antibodies, viral isolation in BHK-21 cells, real-time RT-PCR, conventional PCR, and sequence analysis. While the rabies case reported by Pereira et al. [28] in Brazil provided valuable insights, the incorporation of molecular diagnostics in our study enhances the precision of the findings and offers further information on the circulation of rabies variants among different hosts in the region. Recent studies by Caraballo et al. [34] have demonstrated host shifts in rabies virus variant 2, including its spread to the crab-eating fox (*Cerdocyon thous*), underscoring the complex dynamics of rabies virus evolution and transmission. These findings emphasize the need for similar genetic and ecological approaches to elucidate the dynamics of the prevalent variants in the region and understand their impact on domestic and wild animal populations.

Land use changes, including urbanization and agricultural intensification, have significantly altered the dietary resources available to wildlife [35]. In Latin America, deforestation driven by agricultural and livestock expansion has facilitated the geographic spread of *D. rotundus*, an obligate hematophagous species ranging from Mexico to northern Argentina. Understanding how environmental changes influence the feeding behavior of *D. rotundus* is essential for predicting potential impacts on disease ecology, public health, and ecosystem dynamics. Johnson et al. [36] documented that the intensification of livestock farming has created a stable and abundant blood source, contributing to population growth and establishing *D. rotundus* as the primary sylvatic reservoir for the rabies virus in the region.

In our case study, the infection was attributed to transmission by *D. rotundus*, as evidenced by the proximity of known roosts and direct observations of bat activity at the Guaycolec Wildlife Station, suggesting that the tapir contracted the virus through a bite from an infected bat. In the province of Formosa, 15 cases of rabies in domestic cattle caused by *D. rotundus* were confirmed between 2011 and 2020, showing an increasing trend in cases for that period in Argentina [37]. This scenario is consistent with findings from Pereira et al. [28] in Brazil, where the case was documented in a region neighboring a protected environmental area, as well as close to agricultural lands and urban settings, which provided suitable habitats and nutritional resources for rabies reservoirs.

The province of Formosa in Argentina is characterized by a warm climate, with maximum temperatures ranging between 21.1 °C in winter and 32.5 °C in summer, and experiences one of its peak precipitation periods in April, fostering an environment conducive to bat activity. Temperature is a relevant factor in modulating bat behavior, particularly in response to resource fragmentation. Recent studies have shown that temperatures influence the duration of foraging flights, which affects the ability of *D. rotundus* to adjust its foraging range in response to the fragmentation of resources [38,39]. The rabies case in the tapir reported here occurred in April 2024, during the rainy season, and after a significant heatwave in March, with temperatures reaching 40.3 °C. While no consistent seasonal epidemiological trends for rabies have been observed in the region [14], and it is unlikely that this event is directly associated with seasonal climatic factors; further year-round studies are warranted to explore the potential influence of climatic conditions on rabies dynamics in this area, particularly in relation to increasing land use changes.

Data on rabies exposure in tapirs are currently limited. Aside from a single seropositive tapir documented in São Paulo [40], serological studies of 125 wild tapirs across three Brazilian biomes did not reveal antibody titers indicative of ongoing viral circulation [36]. This underscores the need for further research to elucidate the epidemiology of rabies in lowland tapirs. Although previous studies have indicated a low intrinsic susceptibility of tapirs to rabies [41,42] and estimated only a 0.15% probability of transmission through bites [43], the increasing reports of interactions between hematophagous bats and tapirs in natural habitats [43,44] suggest that bat-associated variants may pose a greater epidemiological risk than previously anticipated.

These observations would lead us to review the role of wild herbivores in rabies ecology. The infrequent detection of rabies in tapirs and other wild herbivores may be attributed to both low intrinsic susceptibility and limitations in surveillance. Given the increasing evidence of interactions between wild herbivores and vampire bats in degraded environments, it is indispensable to strengthen monitoring systems to accurately assess transmission risk and inform targeted conservation and public health interventions in high-risk areas. Our findings are particularly relevant in the context of rabies epidemiology in Argentina, where multiple antigenic variants have been linked to various reservoir species, and bat-mediated transmission is a well-established phenomenon. This case also aligns with concerns raised by Ventura et al. [45] in Brazil regarding rabies threats to wild herbivores. Although the disease is predominantly documented in cattle and horses, its detection in wild herbivores presents a broader epidemiological challenge, necessitating the inclusion of these species in surveillance and prevention strategies. In this context, the rabies vaccination of susceptible captive species housed in enclosures accessible to other animals in wildlife stations should be considered, including tapirs. In our case, SENASA’s local office has implemented comprehensive vaccination campaigns and intensified epidemiological surveillance, not only for tapirs but also for other potential mammalian reservoirs (pumas, monkeys, and deer). Additionally, standardized measures for managing local bat populations have been adopted [13].

Finally, the detection of rabies in a captive tapir reconfirms the species’ susceptibility to the virus, as previously described by Pereira et al. [28], exacerbating existing threats stemming from habitat loss, fragmentation, and the expansion of human activities. This finding reinforces the importance of understanding transmission risk among species at the human–wildlife interface, a concern that is heightened in environments where multiple susceptible species coexist. Strengthening integrated disease surveillance in South America is important for identifying emerging risks, protecting biodiversity, and mitigating the economic and ecological consequences of rabies in domestic and wild herbivores.

Within the One Health framework [4,46], it is essential to enhance surveillance strategies and education to raise awareness of rabies when neurological symptoms are observed in wildlife. Furthermore, ongoing efforts to geolocate and study bat roosts, assess population dynamics, and develop specific intervention strategies are relevant. Investigating the genetic diversity and evolution of rabies virus variants through advanced molecular techniques, as well as analyzing interactions between humans and wildlife—particularly the role of different bat species in transmission—will be critical. Ultimately, exploring the impact of environmental changes and human activities on the emergence and spread of the disease will contribute to a more comprehensive understanding of rabies dynamics.

## 5. Conclusions

This study presents the first confirmed case of rabies in a lowland tapir in Argentina, highlighting the urgent need to expand surveillance efforts to include non-traditional hosts [47]. The identification of antigenic variant 3 in a species not typically associated with rabies underscores the complex epidemiological dynamics of the virus and its potential for spreading in changing environments. Although tapirs have historically been considered of have low susceptibility, increasing interactions with vampire bats in degraded landscapes suggest that this risk may currently be underestimated.

The implications of this case emphasize the necessity of an integrated One Health approach to disease monitoring and management. The detection of rabies in a captive tapir raises concerns about interspecies transmission at the wildlife–livestock–human interface, particularly in regions where natural populations of vampire bats are being heavily impacted by human activities. In this context, strengthening surveillance strategies and enhancing diagnostic capabilities will be essential for mitigating future outbreaks and understanding the epidemiology of rabies in diverse ecosystems.

Furthermore, this study highlights critical knowledge gaps in rabies ecology, particularly regarding wild herbivores as potential hosts, underscoring the need for future research on genetic and ecological studies of rabies virus variants and the effects of environmental changes on host-pathogen dynamics to inform effective conservation strategies and safeguard public health in rabies-endemic regions.

## Figures and Tables

**Figure 1 viruses-17-00570-f001:**
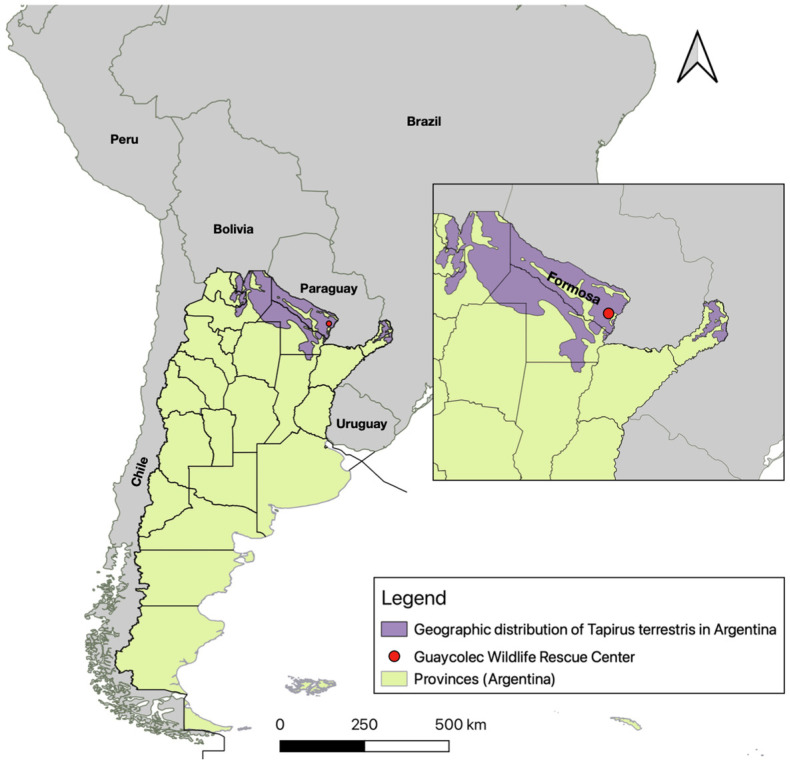
Map of Argentina showing its provinces, neighboring countries, and the location of the Guaycolec Wildlife Rescue Center in Formosa (25°58′55.1″ S, 58°09′42.4″ W). The map also includes the distribution of *Tapirus terrestris* in Argentina [19]. The northeastern region of the country is expanded for a more detailed view.

**Figure 2 viruses-17-00570-f002:**
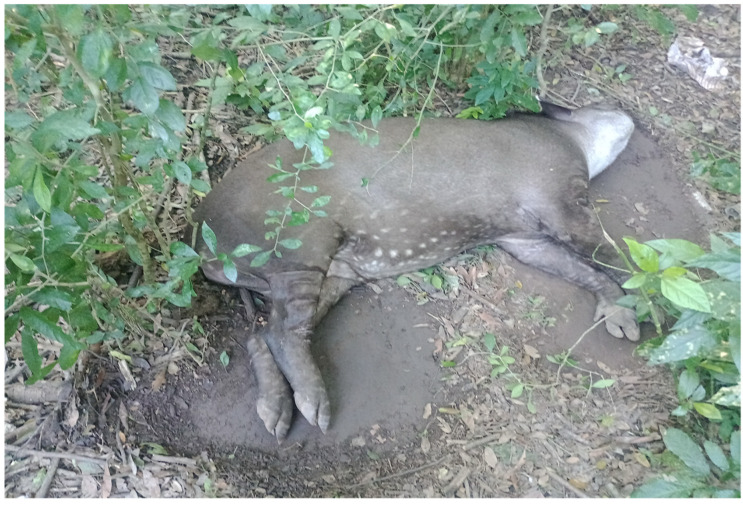
Tapir specimen found in lateral recumbency at the Guaycolec Wildlife Station (Formosa, Argentina), showing signs of paddling on the ground.

**Figure 3 viruses-17-00570-f003:**
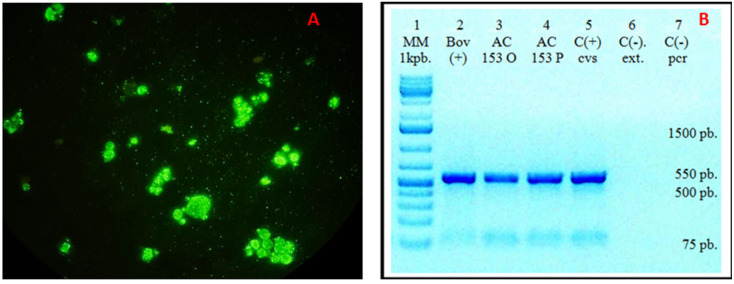
(**A**). Direct immunofluorescence (DIF) of the isolate in BHK-21 cells showing rabies virus-specific fluorescence. (**B**). Conventional PCR detection of rabies in brain samples using primers 3S and 304. The expected 550-bp amplicon was obtained in all samples previously confirmed positive by real-time RT-PCR. Lane 1: Molecular weight marker; Lane 2: Positive bovine sample; Lane 3: AC 153 O (corresponding to AC 153 original-tapir protocol); Lane 4: AC 153 P (corresponding to AC 153 passage-tapir protocol); Lane 5: Positive rabies control; Lane 6: Negative extraction control; Lane 7: Negative amplification control.

**Figure 4 viruses-17-00570-f004:**
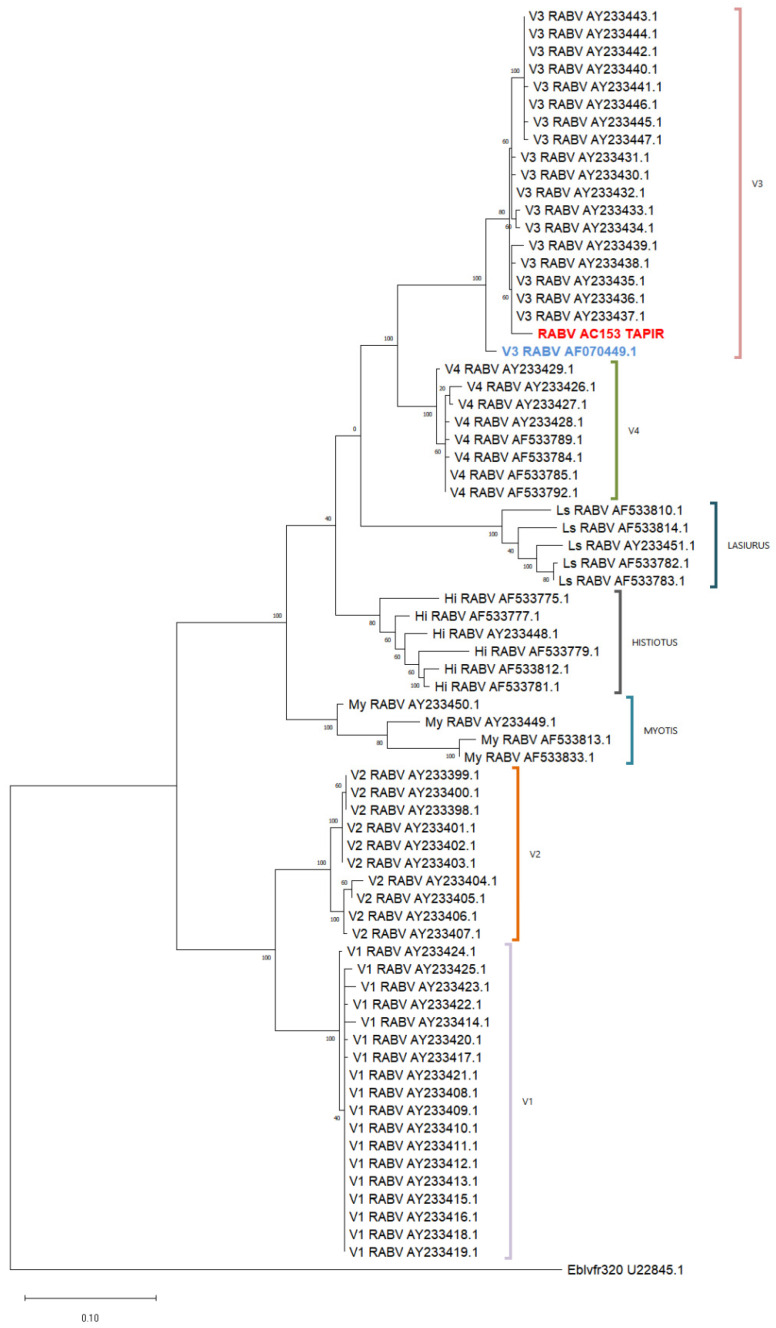
Phylogenetic tree using 320-bp rabies virus sequences. The analysis was conducted using the Maximum Likelihood method and the Tamura–Nei model. The tree with the highest log likelihood (−3999.53) is shown. Sequences are identified by their GenBank accession numbers. The tapir sample is highlighted in red and the vampire bat rabies virus sequence in blue. (My: *Myotis* spp.; Hi: *Histiotus*; Ls: *Lasiurus cinereus*).

**Figure 5 viruses-17-00570-f005:**
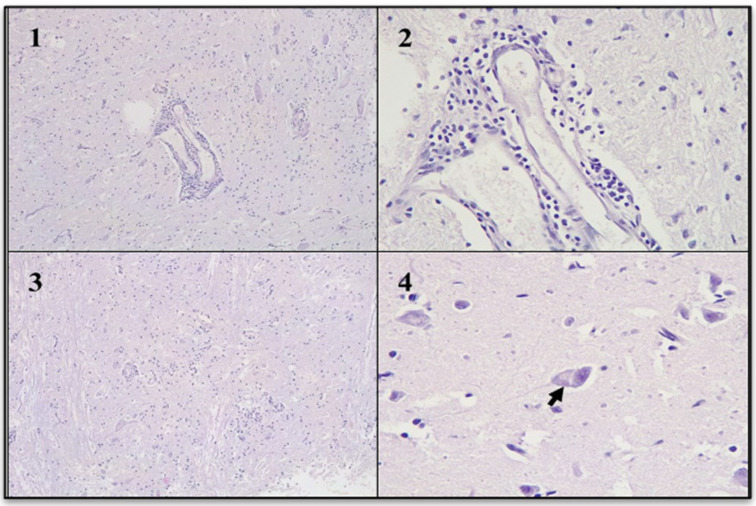
Non-suppurative encephalitis (Hematoxylin and eosin stain). (**1**): Gray matter with perivascular cuffs. (**2**): Perivascular cuffs were composed mainly of lymphocytes. (**3**): Gray matter with focal gliosis. (**4**): Neuronal body with vacuolation of the cytoplasm and intracytoplasmic acidophilic inclusion body (arrow).

## Data Availability

The original contributions presented in this study are included in the article. Further inquiries can be directed to the corresponding author.

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
