# Peer review of "First Report of Paralytic Rabies in a Lowland Tapir (Tapirus terrestris) in Argentina"

_viruses, 2025, doi:10.3390/v17040570_

Round 1
Reviewer 1 Report
Comments and Suggestions for Authors
Nearly 60,000 humans are estimated to die annually from rabies around the world. While this zoonotic infection remains a significant global public health concern, it also presents a considerable conservation challenge. Here, the authors of the manuscript report a case of paralytic rabies in a lowland tapir (Tapirus terrestris) at a wildlife station in Argentina, contributing to the understanding of rabies epidemiology. Clinical signs, necropsy findings and laboratory detection are described in detail. The causative agent was identified as vampire-bite mediated RABV, antigenic variant 3. The manuscript is rather well written in a readable way. It has merit, even though it is only a single case description. Conclusions drawn are reasonable.
Minor comments: 1) The order of figures in the text, which is now Figure 2, Figure 4, and Figure 3, should be corrected, 2) Greater details are needed in the legend of Figure 3, 3) The authors should consider adding a video with clinical signs in the tapir, if available, as supplementary material, 4) Conclusions should, in my opinion, also include a recommendation to vaccinate captive mammals, like tapirs, against rabies in wildlife stations.
Author Response
Response to Reviewer 1:
We sincerely appreciate your positive feedback and constructive comments on our manuscript. Your insights contribute significantly to enhancing the clarity and impact of our work. Below, we address each of your minor comments:
Comment 1: The order of figures in the text, which is now Figure 2, Figure 4, and Figure 3, should be corrected.
Response: We have reordered the figures as requested, including the changes made to incorporate additional figures based on feedback from other reviewers.
Comment 2: Greater details are needed in the legend of Figure 3.
Response: We have expanded the legend of Figure 3 (now Fig 5) to provide greater detail on the histopathological findings in the brain, including descriptions of the lesions observed and relevant histological characteristics. In addition, we removed the second paragraph of the Histopathological Evaluation and replaced it with the morphological diagnosis (lines 283-291).
Comment 3: The authors should consider adding a video with clinical signs in the tapir, if available, as supplementary material.
Response: We appreciate the reviewer’s suggestion. Unfortunately, we do not have video recordings of the clinical signs in the tapir.
Comment 4: Conclusions should, in my opinion, also include a recommendation to vaccinate captive mammals, like tapirs, against rabies in wildlife stations.
Response: We appreciate this important suggestion. In the original manuscript, we discussed vaccination of captive animals, as one of the measures taken in response to this case. In the revised version, we have now explicitly included the recommendation to vaccinate captive tapirs, against rabies in wildlife stations (lines 370-372).

Reviewer 2 Report
Comments and Suggestions for Authors
The paper is very well written. There are a few important points that need to be addressed, but otherwise it is a good start. Understanding the ecological factors that increase the likelihood of pathogens infecting and establishing themselves in a new host is an important area of research. Although this is the first confirmed case of rabies in an Argentinean lowland tapir, the major scientific advantage of this work is the advanced laboratory and molecular diagnosis and sequencing compared to a case of rabies in a tapir in Sao Paulo, Brazil. Considering that these results are the most important, the methods and sequencing results are certainly described too briefly.
2.5. Molecular Diagnosis and Sequencing
- Specify the primer sets used for both qPCR and conventional PCR, either in the text or in a table, with appropriate references.
- For conventional PCR, provide the temperature cycling conditions along with a relevant citation.
- Expand on the sequence analysis by detailing the number of nucleotides before and after editing. Additionally, specify the reference sequence used for comparison.
- If the sequence was submitted to GenBank, include the accession number in the text.
- Clarify how relevant sequences were retrieved: What filtering criteria were applied (country of origin, host)? This info should be added to a tree.
- Describe the method used for sequence alignment in the construction of the phylogenetic tree.
3.3.2. Molecular Diagnosis and Sequencing
- The statement “The sequences were edited using BioEdit v7.2.5, and phylogenetic analysis was conducted using MEGA v11.” does not represent a result. It should be moved to the Methods section.
- Regarding the statement “Alignment with a reference database confirmed that the samples belonged to antigenic variant 3 (Figure 2).”—please specify how the alignment was performed. What reference database was used? Did you made your own database? Based on what criteria? If the data base was retrieved online provide a citation.
- Report the percentage of similarity observed in the alignment.
- Were there any nucleotide variations observed within the sequences?
- Did the alignment reveal any novel or distinguishing mutations?
- How do these sequences compare to previously reported variant 3 sequences?
- Was any evolutionary distance or genetic divergence calculated?
In addition, the authors of the paper need to take better account of the epidemiology of rabies in the region, especially when outbreaks vary greatly (difference between rainy and dry seasons). It is important to specify the exact timing of the outbreak and explain what this means for their conclusions. In this context, it would also be good to mark the geographical location of the Guaycolec wildlife centre in Formosa, Argentina, on the map and compare it with other areas in Argentina, Paraguay and Brazil.
Rabies virus gene sequencing is a method that can provide high-resolution information to distinguish rabies virus variants and help develop a sound control strategy where the sequencing data can be combined with other surveillance data to determine the geographical limits of virus circulation.
Depending on the changes made in the "Result" section, a significantly adapted discussion can also be expected.

Author Response
Response to Reviewer 2:
Comment 1: The paper is very well written. There are a few important points that need to be addressed, but otherwise it is a good start. Understanding the ecological factors that increase the likelihood of pathogens infecting and establishing themselves in a new host is an important area of research. Although this is the first confirmed case of rabies in an Argentinean lowland tapir, the major scientific advantage of this work is the advanced laboratory and molecular diagnosis and sequencing compared to a case of rabies in a tapir in Sao Paulo, Brazil. Considering that these results are the most important, the methods and sequencing results are certainly described too briefly.
Response: Thank you for your thorough review and the positive comments regarding our manuscript. We appreciate your suggestions for improvement. Below, we address your specific points:
2.5 Molecular Diagnosis and Sequencing:
Comment 2: Specify the primer sets used for both qPCR and conventional PCR, either in the text or in a table, with appropriate references.
Response: We have now included detailed information regarding the primers used for both qPCR and conventional PCR in the manuscript (lines 173-179).
Comment 3: For conventional PCR, provide the temperature cycling conditions along with a relevant citation.
Response: We have now incorporated the temperature cycling conditions for conventional PCR in the revised manuscript, along with an appropriate citation (lines 180-185).
Comment 4: Expand on the sequence analysis by detailing the number of nucleotides before and after editing. Additionally, specify the reference sequence used for comparison.
Response: While we appreciate your suggestion, we believe that this information may not be directly relevant to the present publication. Therefore, we have opted to keep the text unchanged. However, we added length information of the sequences used for phylogenetic analysis at the bottom of Figure 3. Moreover, we are also happy to provide additional details for clarity. Sequence analysis was performed using BioEdit software. The original and edited sequences (the outermost regions that do not present a good electropherogram were removed) were: for the Forward sequence 549 and 452 bp respectively and for the Reverse sequence 470 and 348 bp. The manual alignment of the Fw sequence and the complementary Rv gave a consensus sequence of 458 bp. Once the consensus sequence was obtained, a BLAST (https://blast.ncbi.nlm.nih.gov/Blast.cgi) was performed on the National Center for Biotechnology Information (NCBI) website to identify the similarity with other characterized sequences on the GeneBank database. No reference sequence was used.
Comment 5: If the sequence was submitted to GenBank, include the accession number in the text.
Response: The sequence has not yet been submitted to the GenBank database, but we will do so shortly. Once the accession number is assigned, we will include it accordingly.
Comment 6: Clarify how relevant sequences were retrieved: What filtering criteria were applied (country of origin, host)?
Response: We have incorporated into the Materials and Methods section (2.5) the details regarding the selection criteria used for the construction of the phylogenetic tree (lines 194-200).
Comment 7: Describe the method used for sequence alignment in the construction of the phylogenetic tree.
Response: We have added the requested information to the Materials and Methods section (lines 194-200).
3.2.2 Molecular Diagnosis and Sequencing:
Comment 8: The statement “The sequences were edited using BioEdit v7.2.5, and phylogenetic analysis was conducted using MEGA v11.” does not represent a result. It should be moved to the Methods section.
Response: In the revised version, we have removed this statement from the Results section and integrated the relevant information into the Methods section, where these details were already mentioned.
Comment 9: Regarding the statement “Alignment with a reference database confirmed that the samples belonged to antigenic variant 3 (Figure 2).”—please specify how the alignment was performed. What reference database was used? Did you made your own database? Based on what criteria? If the data base was retrieved online provide a citation.
Response: The requested information has been incorporated into the text (lines 254-258).
Comment 10: Report the percentage of similarity observed in the alignment.
Response: The observed similarity with the closest phylogenetic sequence in the tree is 96.57% (AY233437.1). This information is reflected in the phylogenetic tree, which was adjusted accordingly to better represent it (Figure 4).
Comment 11: Were there any nucleotide variations observed within the sequences?
Response: The primary focus of this publication is the presentation of the finding in Tapir. A more detailed analysis of the sequences, including any nucleotide variations, is currently being prepared for a forthcoming publication.
Comment 12: How do these sequences compare to previously reported variant 3 sequences?
Response: This publication primarily focuses on the tapir finding. A detailed comparison of these sequences with previously reported variant 3 sequences is being addressed in an upcoming publication.
Comment 13: Was any evolutionary distance or genetic divergence calculated?
Response: No evolutionary distances or genetic divergences were calculated for this analysis. However, we have updated the format of the phylogenetic tree to ensure it can be more clearly visualized graphically.
Comment 14: In addition, the authors of the paper need to take better account of the epidemiology of rabies in the region, especially when outbreaks vary greatly (difference between rainy and dry seasons). It is important to specify the exact timing of the outbreak and explain what this means for their conclusions. In this context, it would also be good to mark the geographical location of the Guaycolec wildlife centre in Formosa, Argentina, on the map and compare it with other areas in Argentina, Paraguay and Brazil.
Response: In response to your request, we have provided additional information on the timing of the rabies case in the tapir, including a more detailed description of the environmental and climatic context in Formosa (lines 117-124). Also, we have added a paragraph discussing the role of seasonal factors in rabies dynamics and the potential influence of climate conditions on rabies transmission (lines 335-348), as well as the situation in the study area, where no consistent seasonal trends in outbreaks have been observed. Additionally, we have included a figure (Figure 1) with a map marking the geographical location of the Guaycolec Wildlife Center in Argentina.
Comment 15: Rabies virus gene sequencing is a method that can provide high-resolution information to distinguish rabies virus variants and help develop a sound control strategy where the sequencing data can be combined with other surveillance data to determine the geographical limits of virus circulation.
Response: Thank you for highlighting the importance of rabies virus gene sequencing. In the revised version of the manuscript, we have incorporated all the requested information to enhance the understanding of our case report.
Comment 16: Depending on the changes made in the "Result" section, a significantly adapted discussion can also be expected.
Response: We have carefully revised the updated Results section and made corresponding adjustments to the Discussion.

Reviewer 3 Report
Comments and Suggestions for Authors
The authors document the first case of rabies in a Lowland Tapir within Argentina, supported by multiple diagnostic tests. While the finding is interesting, several methodological and reporting issues require clarification:
- Despite the collection of blood samples (line 135), the results of subsequent analyses are not presented.
- The necessity and ethical approval for the mouse inoculation experiment (line 147) should be addressed.
- The methodology for conventional PCR (line 160) is not referenced. A citation is necessary.
- The reported PCR product size is 550bp, yet the sequences used in the alignment presented in Fig. 4 appear to be shorter. The authors must clarify the process by which the alignment was generated.
- Given the hypothesis that vampire bats are the source of the rabies virus, the inclusion and clear identification of vampire bat rabies virus sequences in Fig. 4 would be good. If these sequences are present, they should be explicitly labelled. If not, their addition is recommended to support the authors' claim.
Author Response
Response to Reviewer 3:
We appreciate your careful review and valuable comments on our manuscript. Your insights are essential for improving the quality and rigor of our work. Below, we address your methodological and reporting concerns:
Comment 1: Despite the collection of blood samples (line 135), the results of subsequent analyses are not presented.
Response: We have removed the statement regarding the blood sample analyses. Given that the animal was dehydrated, the values obtained were not deemed reliable for presentation in the manuscript.
Comment 2: The necessity and ethical approval for the mouse inoculation experiment should be addressed.
Response: We have added a statement regarding the necessity and ethical approval for the mouse inoculation experiment: "COMITÉ INSTITUCIONAL PARA EL CUIDADO Y USO DE ANIMALES DE EXPERIMENTACIÓN (CICUAE)- DGLyCT-SENASA, dated April 25, 2024, protocol 2/24 (lines 426-428).
Comment 3: The methodology for conventional PCR (line 160) is not referenced. A citation is necessary.
Response: We have now included the appropriate citation for the conventional PCR methodology and have expanded the details of the methodology in the revised version of the manuscript (lines 171-185).
Comment 4: The reported PCR product size is 550bp, yet the sequences used in the alignment presented in Fig. appear to be shorter. The authors must clarify the process by which the alignment was generated.
Response: Thank you for your observation. The figure has been updated for clarity and is now referred to as Fig. 4.
Comment 5: Given the hypothesis that vampire bats are the source of the rabies virus, the inclusion and clear identification of vampire bat rabies virus sequences in Fig. 4 would be good. If these sequences are present, they should be explicitly labelled. If not, their addition is recommended to support the authors' claim.
Response: The sequence V3 RABV AF070449.1 corresponds to the common vampire bat (Desmodus rotundus). This sequence has been included in the analysis, and it was highlighted in order to be clearly identified in Fig. 4.

Round 2
Reviewer 2 Report
Comments and Suggestions for Authors
After the authors had accepted all the recommendations for improving the article, they improved it considerably so that it could be published in Viruses.